# Learning from others' mistakes: Finetuning machine translation models with span-level error annotations

## Abstract

Despite growing interest in incorporating feedback to improve language models, most efforts focus only on sequence-level annotations. In this work, we explore the potential of utilizing fine-grained span-level annotations from offline datasets to improve model quality. We develop a simple finetuning algorithm, called Training with Annotations (TWA), to directly train machine translation models on such annotated data. TWA utilizes targeted span-level error information while also flexibly learning what to penalize within a span. Moreover, TWA considers the overall trajectory of a sequence when deciding which non-error spans to utilize as positive signals. Experiments on English-German and Chinese-English machine translation show that TWA outperforms baselines such as Supervised FineTuning on sequences filtered for quality and Direct Preference Optimization on pairs constructed from the same data.

## 1 Introduction

Language models have advanced to the point where it is often difficult to improve them substantially via supervised finetuning on high-quality human-written examples alone; instead, recent efforts to improve language model or sequence-to-sequence model performance have largely relied on annotations of model generations, from preferences to per-sequence scores (Bai et al., 2022; Ethayarajh et al., 2022; Lambert et al., 2023; Kopf et al., 2023). Such data, coupled with techniques to learn from it (Christiano et al., 2017; Rafailov et al., 2023; Gulcehre et al., 2023; Dong et al., 2023), have yielded impressive results for many top language models.

Most efforts, however, consider only sequence-level labels, usually in the form of a scalar score assigned to the entire output. In contrast, this work investigates the potential of using fine-grained span-level annotations from offline datasets to enhance language model training. Unlike sequence-level annotations, span-level annotations provide information about specific segments within a sequence, offering more detailed information for model learning. Moreover, in many situations, collecting fine-grained information is similar effort to collecting sequence-level labels (**?**), making the former a practical form of data for improving model performance given a method that can take advantage of the information.

To explore the potential of fine-grained annotations, we focus on the Multidimensional Quality Metrics (MQM) data from previous Workshop on Machine Translation (WMT) Shared Tasks (Freitag et al., 2021a). This data, used to evaluate the quality of machine translation systems, contains span-level annotations of the errors present in a given translation as well as their category (e.g., fluency, accuracy) and severity (e.g., major and minor). While MQM data has previously been used to develop auxiliary reward or metrics models (Juraska et al., 2023; Rei et al., 2022), it has not been directly employed for training machine translation (MT) models.

To directly utilize these translations and their span-level annotations to finetune an MT model, we introduce a new algorithm called Training with Annotations (TWA). TWA utilizes span-level information from the annotations to treat error and non-error spans differently. For error spans, the TWA loss seeks to decrease the probability of the span given the context while allowing the model to learn which tokens in the span to penalize to do so. For non-error tokens, TWA takes into account

Figure 1: Overview of Training with Annotations (TWA). TWA proceeds by tokenizing the output text and its annotations. Then, a weighted span-level unlikelihood loss is applied to each error span to allow the model to learn what parts of the error span to penalize and non-error tokens following an error span are ignored as they are off-trajectory. All other tokens (i.e., non-error tokens preceding an error span) are trained with cross entropy loss.

the overall sequence trajectory when deciding which spans should be treated as positive signals. A high-level summary of TWA can be found in Figure 1.

Experiments on English-German and Chinese-English machine translation demonstrate that TWA yields significant improvements over baselines which either do not consider annotation information or only utilize the information at the sequence level. Specifically, TWA can outperform methods such as supervised finetuning on sequences filtered for quality and Direct Preference Optimization (DPO) on preference pairs constructed from the same data. These results highlight the effectiveness of taking advantage of span-level annotations to improve model performance.

First, we describe the MQM data and the information provided in the span-level annotations (Section 2). Then, we discuss existing work which either utilizes the MQM data or the fine-grained annotations (Section 3). Then, we introduce our method, Training with Annotations (TWA), in Section 4. We outline our experimental setup in Section 5 and present the results in Section 6. Finally, we conclude with a discussion of our findings and future work in Section 7.

## 2   MQM DATA

Each year, the Workshop on Machine Translation (WMT) hosts a shared task competition to assess general machine translation capabilities across different domains and genres. Submitted MT systems are scored and evaluated by humans, with top systems annotated via the Multidimensional Quality Metrics (MQM) scheme (Freitag et al., 2021b; Rei et al., 2022). Namely, given the source text and MT output, professional translators annotate any error spans in the output translation. Each error span is annotated with the category of the error as well as the severity of the error. Each error span is assigned a score of 25 for a non-translation, 5 for a major error, 0.1 for a minor punctuation error, and 1 for any other minor error. The overall MQM score of an example sequence is the sum of the MQM scores of the annotated error spans in the sequence.

MQM annotations have been used to evaluate MT systems, as described above, but not as additional training signal to finetune MT models. Utilizing these annotations during training requires developing a method that can take this information into account. We describe our proposed method, Training with Annotations, in Section 4.

## 3   RELATED WORK

**Utilizing MQM data.**   TWA is the first method to use span-level MQM data to directly finetune machine translation models, but there exist other methods which also utilize sequence-level MQM data indirectly. Namely, existing automated metrics in machine translation such as MetricX (Juraska et al., 2023) utilize MQM scores as labels for training data, so methods which utilize these neural-based automated metrics indirectly benefit from MQM data. Such approaches include QE reranking (Fernandes et al., 2022) or MBR decoding (Freitag et al., 2022) with neural quality metrics. Both

methods can be used in tandem with TWA, as one could always decode a TWA-trained model with either of these approaches. One could also use the results of such decoding methods to directly fine-tune a model, commonly known as MBR or QE finetuning (Finkelstein & Freitag, 2024). However, given the models powering automated metrics such as Metric-X are trained on multiple sources of data beyond that of MQM data alone, MBR and QE finetuning are not directly comparable with TWA.

**Utilizing fine-grained annotations.** There exist other methods which consider fine-grained annotations, but they consider a different setting than TWA. Fine-grained RLHF (FG-RLHF) (Wu et al., 2023) adapts RLHF to reward models which provide finer-grained feedback than a single sequence-level score. Similar to our work, Wu et al. (2023) achieve better performance using fine-grained RLHF with span-level rewards than using RLHF with sequence-level rewards. The difference between FG-RLHF and TWA is that the former is a reinforcement learning method that requires an auxiliary fine-grained reward model to annotate model generations online, while the latter is a fine-tuning method that can work directly with offline annotated data without the need for additional models during training. The performance of FG-RLHF depends on the quality of the fine-grained annotator model, which can be difficult to develop (see Pang et al. (2023) and Appendix C). Moreover, accuracy of the annotations aside, a reinforcement learning approach which only takes into account online data misses out on the opportunity to learn from offline examples themselves, not just their annotations.

Next, Targeted Negative Training (TNT) (Zhang et al., 2024) is a method for training on token-level annotations of negative examples, but its motivation is to achieve a targeted update, i.e., reducing unwanted behavior while minimally changing the model otherwise. TWA, on the other hand, is not concerned with making precise updates but rather improving overall quality as much as possible. Finally, FUDGE (Yang & Klein, 2021) is an alternative decoding technique which utilizes a token-level auxiliary reward model to sample from the model conditioned on a given attribute $a$; namely, given reward model which approximates $p(a|y_{\leq t}, x)$, FUDGE samples from $p(y_t|y_{<t}, x, a)$ using the original model $p(y_t|y_{<t}, x)$ and reward model $p(a|y_{<=t}, x)$. TWA, on the other hand, is a finetuning-based approach that does not alter the test-time behavior of the model and does not require an auxiliary reward model.

## 4 TRAINING WITH ANNOTATIONS

Training with annotations (TWA) is a finetuning algorithm that takes into account example outputs and their span-level error annotations. TWA proceeds as follows: first, the example is tokenized and given weights corresponding to its annotations: tokens which contain any characters within an error span are given a negative weight, and tokens outside an error span are given a non-negative weight. Then, during training, the TWA loss for a given sequence is a sum of the losses from the error spans and the non-error tokens. Below, we describe and motivate the choices for the constituent losses.

### 4.1 HANDLING ERROR SPANS

An annotated error span provides information to the model that such a continuation is undesirable given the preceding context (and thus should be unlikely under the model). To decrease the probability of error spans given their context, TWA utilizes the unlikelihood loss, $-\log(1-p)$. The loss is high when the probability $p$ is high and 0 when $p$ is zero. In Section 6, we consider alternative choices of loss for error tokens and find that the unlikelihood loss outperforms other choices. Moreover, the unlikelihood loss is efficient to compute as it only requires access to the current model being trained.

Applying unlikelihood to each token in an error span may not be desirable, however. Take the output in Figure 1, for example. Imagine the correct translation was "Give me an example of a blessing in adversity", but the submitted translation was "Give me a story about a blessing in disguise", as shown in the figure. Moreover, say the sequence was tokenized in the way shown in the figure, with "disguise" being tokenized into "dis" and "guise". First, even though "disguise" is an inaccurate translation of "adversity", "guise" is perhaps the most reasonable continuation of the sequence given the prefix ends with "blessing in dis". Penalizing "guise" given its prefix does not necessarily reflect the intention of the error span; rather, it is probably more appropriate to assign a low probability to

"dis" given its prefix while maintaining a high probability for "guise" given a prefix ending in "a blessing in dis". Second, the error annotation around "a story about" does not necessarily mean that the article 'a' and the preposition "about" should be assigned a low probability given their prefixes. The above examples are just a few instances of the broader idea that not all tokens in an error span should be penalized.

Given these examples and others, one might be able to come up with a series of heuristics to transform the resulting span-level errors into corresponding token-level losses. However, as is common in natural language, manually creating rules can be difficult and error-prone (whether due to low precision or recall). Instead, we choose to let the model learn what to penalize within a span by utilizing a span-level unlikelihood term instead of a token-level one. We additionally take into account the severity of the error by scaling the loss by the absolute value of the severity weight $w$ assigned to the span, equal to the error span's negative MQM score: -0.1 for minor punctuation, -1 for all other minor errors, and -5 for major errors.[1] The loss for an error span is the following:

$$\mathcal{L}_{\text{TWA}}(\text{error span}) = -|w|\log(1 - p_{\text{span}}) = -|w|\log(1 - \exp\sum_{t \in \text{span}}\log p_t). \tag{1}$$

Rather than forcing the model to push down probability over all tokens in a span given their prefixes, the span-level unlikelihood loss allows the model to learn which tokens to penalize in order to decrease the overall probability of the span.

### 4.2 Handling non-error spans

When the overall quality of the data is high relative to the base model, using supervised finetuning (SFT) to maximize the likelihood of the translations in the data can improve the model. On the other hand, when the overall quality of the data is low relative to the model, SFT can hurt performance, by teaching the model to reproduce errors. Thus, to optimize model quality, most efforts seek to filter out low-quality examples and train just on high-quality ones. However, in reality, there is likely often a spectrum of translation quality even within an example itself. Fine-grained annotations provide extra information about this variation in quality by pinpointing exactly where errors exist. Then, for all other tokens, we can proceed with typical maximum likelihood training via cross entropy loss, without worrying about maximizing the likelihood of errors.

However, all the subsequent tokens after an error are out-of-support since their prefixes contain an error that should be low or zero probability under the intended new model. We call these subsequent tokens *off-trajectory*. Generalization aside, off-trajectory tokens at best are irrelevant to the model distribution and at worst could provide noisy signal. While there is an argument that high-quality off-trajectory tokens could provide signal that generalizes to trajectories the model will actually sample, we find empirically that ignoring these tokens in the overall loss can greatly improve performance in some settings (see Table 4). TWA on non-error spans is thus as follows:

$$\mathcal{L}_{\text{TWA}}(\text{non-error span}) = \begin{cases} 0 \text{ if span after first error} \\ -\log p_{\text{span}} \text{ otherwise.} \end{cases} \tag{2}$$

Note that this is equivalent to employing per-token cross entropy loss on non-error tokens before an error span, as $\log p_{\text{span}} = \sum_{t \in \text{span}}\log p_t$.

### 4.3 Overall method

Combining the insights from the above two sections, we have a simple finetuning algorithm for TWA as depicted in Figure 1. First, we tokenize the output sequence and its corresponding annotations. The latter become weights which are negative values for tokens with characters contained in an annotated error span, zero for all tokens following the first error span, and one for all other non-error tokens. Then, we group tokens into spans based on weight (i.e., all contiguous tokens with the same weight are in the same span) and employ either the TWA error span loss (Equation (1)) or the TWA non-error span loss (Equation (2)). The overall TWA loss for a given sequence is the sum of all the span losses.

---

[1]Under the MQM rating system, some major errors are given a score of -25 (namely those categorized as non-translations), but we use a weight of -5 for these errors as well.

## 5 EXPERIMENTS

### 5.1 DATA

**Pretraining.** We pretrain En→De and Zh→En models using the parallel WMT'23 training data (Kocmi et al., 2023), which consists of 296 million sentence-level examples. For En→De, we additionally construct multi-sentence examples from a subset of this data where the overall documents can be recovered and partitioned into longer blocks than those of individual sentences. The multi-sentence examples have a max length of 1024 tokens, with 512 tokens each for the input source and output target.

**Finetuning.** For both language pairs, we then apply TWA on top of the pretrained model, using MQM data from WMT'20 (Barrault) and WMT'21 (Akhbardeh) for training. In total, the training dataset contains roughly 2,900 and 3,100 source texts, with around 28,000 and 31,000 submission outputs for En→De and Zh→En, respectively (around ten submissions per source on average).

### 5.2 BASE MODEL

For both language pairs (En→De and Zh→En), we use a 602-million-parameter Transformer encoder-decoder architecture implemented in *Pax*[2]. The model has 8 encoder and 8 decoder layers (rather than 6), but otherwise is similar to the *transformer-big* setting in Vaswani et al. (2017), with model dimension of 1024, hidden dimension of 8192, and 16 multi-attention heads. For each language pair, we use a bilingual vocabulary of 32k subword units trained on the WMT'23 training dataset (Kocmi et al., 2023). We pretrain with the standard cross entropy loss.

See Table 1 for a comparison of the quality of our base model relative to the average quality of the WMT'20-'21 submissions, and Table 2 for the range of quality across submissions (best and worst systems). On average, the submissions are higher quality than our starting base model.

See Appendix A for additional statistics between the base model and submissions data, including error token distributions (Figure 3) and histograms per-sequence of quality scores between model generations and data (Figure 4).

Table 1: Quality of original base model and submissions data (all systems in aggregate).

|  | En → De | | Zh → En | |
| --- | --- | --- | --- | --- |
|  | Metric-X ↓ | COMET ↑ | Metric-X ↓ | COMET ↑ |
| base model | 2.132 | 0.406 | 4.529 | 0.326 |
| submissions data | 1.301 | 0.525 | 3.414 | 0.376 |

Table 2: Quality of best and worst system submissions.

|  | En → De | | Zh → En | |
| --- | --- | --- | --- | --- |
|  | Metric-X ↓ | COMET ↑ | Metric-X ↓ | COMET ↑ |
| Best | 0.194 | 0.641 | 2.258 | 0.517 |
| Worst | 2.043 | 0.192 | 3.573 | 0.193 |

### 5.3 BASELINES

We compare TWA with Supervised FineTuning (SFT) and Direct Preference Optimization (DPO) (Rafailov et al., 2023) as baselines. SFT on the MQM annotated data is analogous to distilling the outputs of other MT systems, without taking into account the annotations. DPO is a preference learning algorithm which operates on pairs of responses to the same input given the knowledge that

---

[2]https://github.com/google/paxml

one response in the pair is preferred to another. We construct response pairs for DPO using the sequence-level MQM scores (i.e., the sum of the MQM scores of all the error spans), creating pairs from all combinations of system translations to the same source input where the MQM score is distinct. We arrived at this setting after testing multiple variations; see Appendix B for details. In other words, DPO utilizes the annotations as additional information, but only at a sequence level.

When using both submissions and references for finetuning, we treat references as error-free for TWA and TWA-seq, and treat them as better than all submissions for constructing DPO pairs; the resulting dataset for DPO thus contains all the pairs constructed from submissions only, plus additional (reference, submission) pairs for every submission. We also consider two additional baselines. First, given the quality of the data makes a big difference in the efficacy of SFT, we construct a dataset of only the references and error-free submissions and run SFT on this filtered dataset. We call this baseline Filter + SFT. Second, we also run a sequence-level analogue to TWA, where we apply a sequence-level unlikelihood loss to an output if it contains any error and cross entropy loss otherwise. We call this baseline TWA-seq.

For all the methods, we use a batch size of 8192 (4096 pairs for DPO), a learning rate of 2e-6 with a constant schedule, and no label smoothing. Greedy decoding is used throughout the experiments.

## 5.4 EVALUATION

For evaluation, we use MetricX-23 (Juraska et al., 2023) and COMET-20 (Rei et al., 2020) as quality metrics. MetricX-23 is a reference-based metric which scores a translation based on a reference and a hypothesis, without taking into account the source text. COMET-20 takes into account the source text, hypothesis, and reference translation. Moreover, MetricX-23 has been finetuned on MQM WMT'20-'21 data, while COMET-20 has not. Given their differences, considering both automated quality metrics helps guard against overfitting to the idiosyncrasies of either. Lower is better for MetricX while higher is better for COMET-20; hence, for checkpoint selection, we average the values of MetricX-23 and the negative COMET-20 on the validation set every 500 steps, selecting the checkpoint with the lowest score. Throughout the rest of the paper, we use MetricX and COMET to denote MetricX-23 and COMET-20, respectively.

We use the generalMT2022 test set (Kocmi et al., 2022) as our validation set for checkpoint selection, and report all results on the WMT'23 (Kocmi et al., 2023) test set. The validation set contains roughly $2,000$ and $1,900$ source texts (along with their corresponding reference translations) for En→De and Zh→En, while the test set contains 600 and $2,000$ examples for En→De and Zh→En, respectively. Note that the WMT'23 En→De test set is paragraph-level.

## 6 RESULTS

### 6.1 MAIN RESULTS

First, we compare TWA to the baselines described in Section 5.3. We perform experiments using the submissions data alone, as well as in tandem with the human-written reference translations (one per source). We also report performance clusters based on statistically significant performance differences between pairs. For each language pair and data source (i.e. submissions only vs. submissions+references), we verify whether the measured differences between each system pair is statistically significant via a paired permutation test[3] using 1000 re-sampling runs and a significance level of $p = 0.05$. We then group systems with similar performance by following the clustering procedure from (Freitag et al., 2023). Namely, given significance results (p-values) for all pairs of systems, we assign ranks as follows. Starting with the highest-scoring system, we move down the list of systems in descending order by score, and assign rank 1 to all systems until we encounter the first system that is significantly different from any that have been visited so far in the latter cluster. That system is assigned rank 2, and the process is repeated until all systems have been assigned a rank. This clustering is done independently for each automated metric.

---

[3]Considering each system as its distribution of the MetricX or COMET scores for each source-translation pair, we test how likely a given result between pairs of systems would be if their underlying distribution of scores were the same. In code, we use `scipy.stats.permutation_test(*, statistic=np.mean, permutation_type='samples')`

Table 3 summarizes the results. We find that TWA significantly outpeforms all baselines in En→De translation and is always within the top-performing cluster for all settings. All methods improve quality over the base model, which is in line with the fact that the submissions data are of higher quality overall than the base model's generations. TWA's consistent improvement over SFT suggests that even when the data is overall of better quality than the current model being finetuned (i.e., training on all the data still improves performance), it can still be beneficial to treat some spans differently than others. The fact that sequence-level baselines that take into account negative information (i.e., DPO, TWA-seq) do not necessarily improve performance over SFT highlights the challenge of attribution when utilizing sequence-level information. Namely, both DPO and TWA-seq utilize more information than SFT (i.e., DPO takes into account that one sequence is preferred over another, while TWA-seq knows which sequences have errors and which ones are error-free), but they are not able to effectively utilize this information to gain a systematic improvement over a baseline that ignores this information. These results suggest that even when extra information is available, it is non-trivial to develop a method which can effectively take advantage of this information. TWA, on the other hand, is able to take advantage of span-level annotation information to outperform SFT and Filter+SFT, highlighting the effectiveness of the method. TWA's improvement over Filter + SFT (significant for En→De) demonstrates that it is able to utilize useful signal that is otherwise thrown away with sequence-level filtering.

Overall, TWA is best performing across the board, significantly so over all baselines for En→De and consistently in the rank-1 cluster for Zh→En. For Zh→En submissions only, TWA is in the same cluster as DPO for Metric-X. While DPO may seem better on Metric-X (though not significantly so), it is substantially worse on COMET (less than half the COMET score of TWA), suggesting that DPO has exploited an idiosyncrasy of the Metric-X model without truly improving in overall performance. For Zh→En references and submissions, only TWA and TWA-seq are in the rank-1 cluster for both Metric-X and COMET.

Table 3: Results aggregated by language pair and automatic metric. We also indicate the data sources used for each result. Models with statistically significant performance improvements are grouped in quality clusters. We highlight the best ranked models in bold.

| | Submissions | References | En → De | | Zh → En | |
| --- | --- | --- | --- | --- | --- | --- |
| | | | Metric-X ↓ | COMET ↑ | Metric-X ↓ | COMET ↑ |
| Base model | | | 4.203 | 0.429 | 4.938 | 0.066 |
| SFT | ✓ | | 3.573 [2] | 0.481 [2] | 4.253 [2] | 0.255 [2] |
| DPO | ✓ | | 3.792 [2] | 0.455 [3] | **4.072** [1] | 0.113 [3] |
| TWA | ✓ | | **2.944** [1] | **0.507** [1] | 4.091 [1] | **0.277** [1] |
| SFT | ✓ | ✓ | 3.159 [3] | 0.491 [2] | 4.094 [3] | 0.271 [2] |
| DPO | ✓ | ✓ | 3.564 [4] | 0.442 [3] | 4.063 [2] | 0.113 [3] |
| Filter + SFT | ✓ | ✓ | 2.950 [2] | 0.499 [2] | 4.004 [2] | 0.289 [1] |
| TWA-seq | ✓ | ✓ | 3.158 [3] | 0.485 [2] | 3.993 [1] | 0.284 [1] |
| TWA | ✓ | ✓ | **2.882** [1] | **0.513** [1] | **3.965** [1] | **0.290** [1] |

## 6.2 TWA ABLATIONS

Next, we isolate the effect of the individual components of TWA in Table 4. Starting from the base model, we note first that training on all the submissions (+ SFT on submissions) improves results. Then, given knowledge of span-level errors, the most obvious next step is to treat the tokens with and without errors differently. Absent a method to deal with errors, the most straightforward next step is to include only the non-error tokens in the loss, ignoring the error tokens to prevent the model from maximizing the likelihood of them given their context. We see that this step (+ on non-error tokens only) improves results over training on all error tokens, confirming our hypothesis that training on error tokens negatively contributes to model quality. Then, we incorporate the TWA loss on error spans, whose tokens make up on average 11.0% and 13.6% of the total tokens in a given translation (see Figure 3 for additional statistics on the error- vs. non-error makeup of the data). This results in further improvements, demonstrating that it is possible to improve model quality by learning from

Table 4: A breakdown of the components of TWA and their isolated effect on model quality. Models with statistically significant performance improvements are grouped in quality clusters, and the best ranked scores are shown in bold.

| | En $\to$ De | | Zh $\to$ En | |
|---|---|---|---|---|
| | Metric-X $\downarrow$ | COMET $\uparrow$ | Metric-X $\downarrow$ | COMET $\uparrow$ |
| Base model | 4.203 | 0.429 | 4.938 | 0.066 |
| + SFT on submissions | 3.573 [4] | 0.481 [4] | 4.253 [2] | 0.255 [3] |
| + on non-error tokens only | 3.488 [3] | 0.487 [3] | 4.120 [1] | 0.283 [1] |
| + span-level loss on errors | 3.325 [2] | 0.495 [2] | **4.088** [1] | **0.284** [1] |
| + ignore off-trajectory tokens | **2.944** [1] | **0.507** [1] | 4.091 [1] | 0.277 [2] |

negative information over ignoring errors entirely. Finally, we ignore off-trajectory tokens, which results in substantial gains in En→De but not in Zh→En.

## 6.3 NEGATIVE LOSSES FOR TWA

A key component of TWA is how to utilize error spans as negative information. In Table 5, we compare the unlikelihood loss used in TWA with the negative likelihood loss also on the span level, i.e., $\mathcal{L}_{NL}(\text{span}) = \log p_{\text{span}}$. Table 5 shows that unlikelihood greatly outperforms negative likelihood. This is likely due to the fact that the negative likelihood only grows in its contribution to the loss and corresponding gradient as the probability of an error span goes to zero (i.e., $\lim_{p \to 0} \log p_{\text{span}} = -\infty$ and $\lim_{p \to 0} \frac{\partial}{\partial p} \log p_{\text{span}} = \infty$) and can thus outweigh likelihood terms as the probability of positive spans moves towards 1 (i.e., $\lim_{p \to 1} - \log p_{\text{span}} = 0$ and $\lim_{p \to 1} \frac{\partial}{\partial p} - \log p_{\text{span}} = -1$). In contrast, unlikelihood mirrors the loss and gradient of likelihood as the span probability moves towards the desired result (i.e., $\lim_{p \to 0} - \log(1 - p_{\text{span}}) = 0$ and $\lim_{p \to 0} \frac{\partial}{\partial p} - \log(1 - p_{\text{span}}) = -1$

Table 5: Comparison of negative losses for use on error spans. We compare unlikelihood (UL), the choice in TWA, with negative likelihood (NL).

| | En $\to$ De | | Zh $\to$ En | |
|---|---|---|---|---|
| Loss | Metric-X $\downarrow$ | COMET $\uparrow$ | Metric-X $\downarrow$ | COMET $\uparrow$ |
| UL | 2.944 | 0.507 | 4.091 | 0.277 |
| NL | 3.477 | 0.491 | 4.730 | 0.108 |

## 6.4 ANALYZING TWA

Next, we visualize how TWA changes the model distribution. For each submission output in the training data, we obtain its per-token log probabilities. Moreover, for each token we record its log probability rank under the model relative to all other tokens in the vocabulary. Both can be obtained through a single forward pass. We obtain log probability ranks for both the original base model as well as the TWA-trained model and compute the change in rank for each token from the base model to the TWA-trained model. Note that since the model is decoded via greedy decoding, changes in rank are more indicative of behavior shifts than changes in log probability. We visualize the changes in rank for four different sample training examples in Figure 2. Notably, the configuration of tokens penalized within the error span varies across different samples, demonstrating the flexibility of span-level error loss in enabling the model to learn which tokens to penalize—an outcome that would be challenging to encode manually with a set of heuristics. Quantitatively, we also find that utilizing a span-level error loss substantially outperforms using a token-level loss on each token in a span (3.325/0.495 MetricX/COMET vs. 3.433/0.470 for token-level on En→De submissions only).

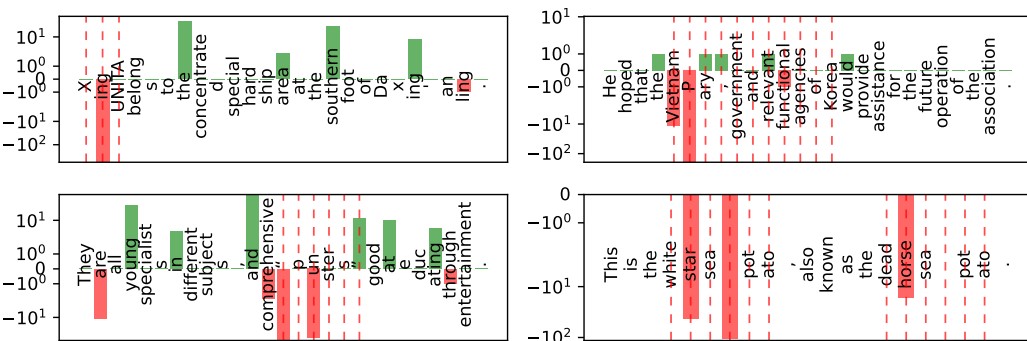

Figure 2: Change in the rank of each token in the vocabulary from the base model to the TWA-trained model. Dashed red lines indicate annotated errors. Red bars show a worsening in rank, while green bars indicate improvement. TWA learns diverse patterns for penalizing specific token conditionals within an error span—patterns that would be challenging to capture with heuristics.

### 6.5 TWA WITH ON-POLICY SAMPLES

While the aforementioned experiments all utilize off-policy data generated from MT systems other than the one being finetuned, next we test the efficacy of TWA in an on-policy setting. Concretely, we obtain MQM annotations of the base model's translations and run TWA with this annotated data. We see substantial improvements in quality, from 4.203/0.429 Metric-X/COMET to 3.710/0.456 Metric-X/COMET. While these improvements from online data are not as large as those with the off-policy data, due to the fact that the submissions data is on average better quality than the base model's translations (see Table 1), the fact that TWA significantly improves over the base model in this setting speaks to the ability of the method to specifically take advantage of annotation information.

## 7 DISCUSSION

In this work, we introduce Training with Annotations (TWA), a method for finetuning a language model on data with span-level error annotations. While most existing efforts have focused on utilizing sequence-level annotations, TWA can take advantage of finer-grained information for more effective learning. Our experiments on English-German and Chinese-English machine translation highlight the performance gains TWA offers compared to methods that focus solely on sequence-level information.

As model capabilities continue to improve, it will be increasingly difficult to rely on the collection or construction of high-quality examples as training signals. In fact, many of the MT system submissions in WMT'24 were found to surpass the quality of human-constructed reference translations, highlighting the need to move beyond demonstration data for improving existing models. MQM annotations of model generations offer a valuable alternative source of information for model training, and TWA unlocks the potential to utilize such rich information directly and simply.

Yet while the experiments focus on MQM data for the task of machine translation, TWA can be used for span-level annotations broadly, paving the way for other applications of fine-grained annotations. While fine-grained information may be more expensive to collect than sequence-level information for some tasks, Wu et al. (2023) find that for long-form question-answering, the time required for humans to annotate span-level errors is comparable to the time required to label the sequence overall. Many other tasks likely fall into this same category: for instance, one needs to locate the hallucination in order to label a sequence as "has hallucination"; similarly, identifying specific spans of bias or misinformation is necessary before assigning a label such as "biased" or "inaccurate".

There exist multiple ways to build upon TWA. One avenue for future work would be to apply TWA in settings beyond machine translation or to language models in general. Another would be to additionally take into account the fine-grained annotation information in other ways—for instance, given

fine-grained information provides a natural ranking of inputs, one could consider directly providing the model with this relative quality information as well. Other interesting questions to investigate include assessing TWA on online data, analyzing the impact of the quality of the generations and annotations on resulting model performance, and exploring the repeated use of TWA for iterative refinement of a model. Finally, the fact that ignoring off-trajectory tokens was highly beneficial in one language pair but not in the other, provides an opportunity to further refine TWA to better handle off-trajectory tokens since the latter might contain additional useful information for training.

In summary, TWA offers a straightforward method to capitalize on existing span-level annotation data as well as a reason to begin collecting span-level information in applications which currently do not. By taking advantage of previously overlooked sources of supervision, methods such as TWA can help unlock new avenues for pushing the frontier of model development.

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

## A    ADDITIONAL DATASET & MODEL STATISTICS

While Table 1 and Table 2 present average quality scores for the system submissions and base model, here we present additional statistics for both.

Table 6: Average length and percentage of error tokens for En-De and Zh-En translation pairs. Standard deviations are shown in parentheses.

|  | Average length (tokens) | Percentage of error tokens |
|---|---|---|
| En $\to$ De | 38.8 (24.6) | 11.0 (19.6) |
| Zh $\to$ En | 40.4 (27.0) | 13.6 (19.2) |

## B    DPO HYPERPARAMETER SWEEPS

To ensure a fair comparison with baseline methods, we test many settings of DPO, varying the construction of the preference pairs and the method for scoring sequences to determined preferred vs. dispreferred in a pair. We set $\beta = 0.1$. Table 7 summarizes the results. As the DPO loss seeks to increase the probability of the preferred sequence relative to its probability under the original model and decrease the probability of the dispreferred sequence relative to its probability under the original model, we first constructed pairs where the reference was always the preferred sequence in a pair. As the dispreferred sequence, we tested using the best submission (by MQM score), worst submission, or all submissions and found that using the worst submission yielded the best results. However, the performance in all these settings paled in comparison to the setting where we constructed as many pairs of distinct score submissions as possible, even without access to the reference data. Adding additional pairs using the reference data improved results further, so we chose this setting for constructing pairs. With this setting, we find that using the sum of the span-level MQM scores performs better than the mean MQM score when both references and all submissions are applied; given that sequence-level MQM scores are generally computed using the sum, we choose it over the mean.

## C    FINE-GRAINED ANNOTATOR MODEL

Here, we consider the endeavor of developing a model to output fine-grained annotations of a sequence. We consider two approaches, direct finetuning and in-context learning (Brown, 2020) with Gemini Pro-1.5 (Team, 2024). For the former, we use the WMT'20-'22 MQM datasets. For the latter, we use the MQM submissions data matching a given source input as in-context examples for annotating a given output translation for that same source. We utilize the following prompt preceding the ICL examples: "You are an annotator for the quality of machine translation. Your task is to identify errors and assess the quality of the translation". We test both approaches on the WMT'23

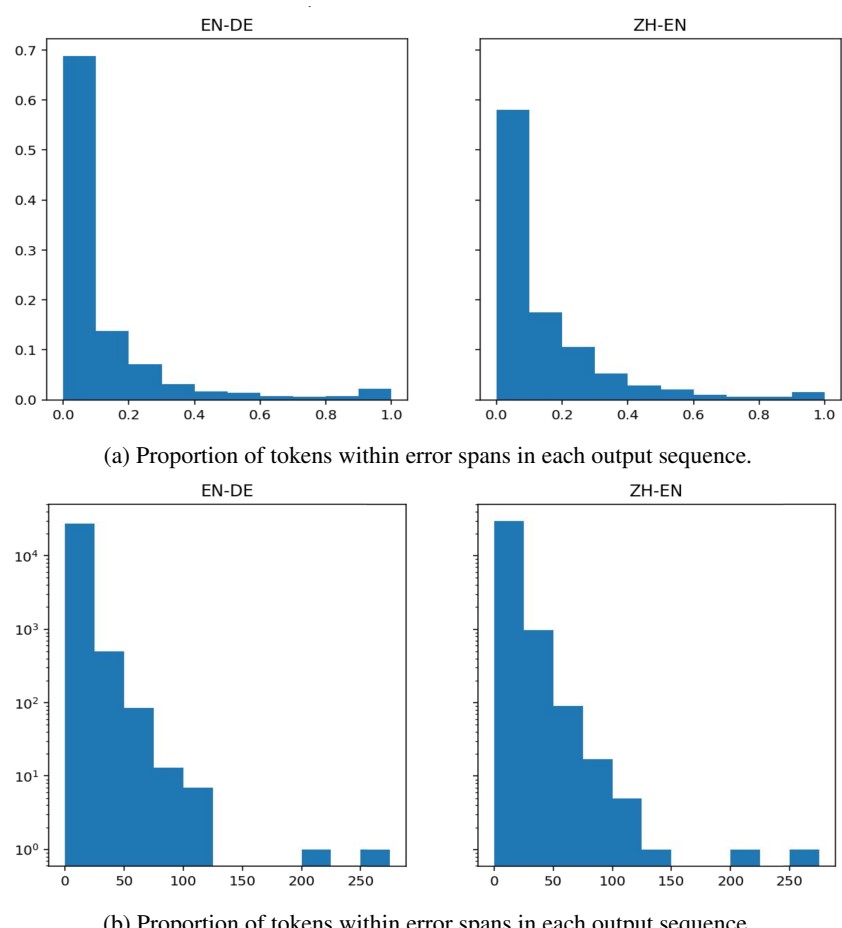

(a) Proportion of tokens within error spans in each output sequence.

(b) Proportion of tokens within error spans in each output sequence.

Figure 3: Histograms of the proportion and number of errors in the training data. Left is En-De, right is Zh-En.

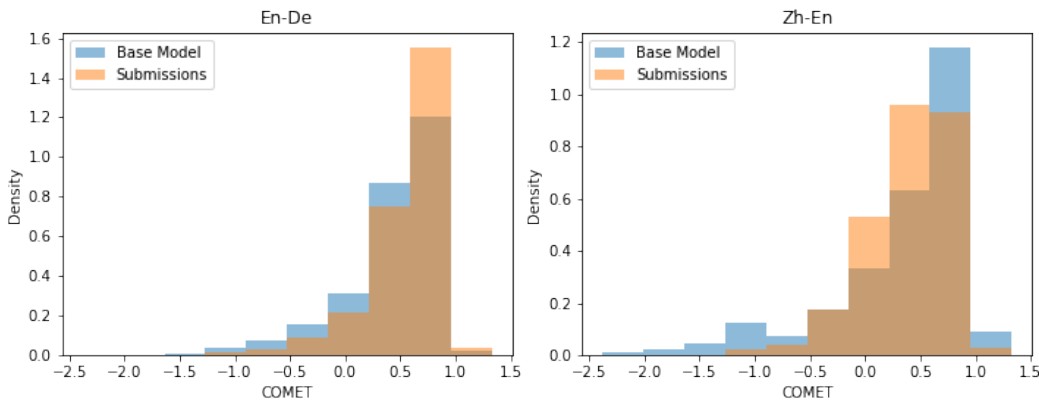

Figure 4: Histogram of COMET scores across the submissions and base model generations. Source inputs come from the training data.

test set and find that the latter (ICL) yields better results than the former (direct finetuning). Thus, we use the latter to annotate our base model generated translations.

Table 7: DPO En→De results (test set) for different configuration settings.

| Setting | preferred | dispreferred | score | Metric-X ↓ | COMET ↑ |
|---------|-----------|--------------|-------|-----------|---------|
| Default | reference | best submission | mean | 6.259 | 0.083 |
| Default | reference | worst submission | mean | 5.540 | 0.174 |
| Default | reference | all submissions | mean | 5.575 | 0.157 |
| Default | all submissions | all submissions | mean | 3.753 | 0.455 |
| Default | reference, all submissions | all submissions | mean | 3.739 | 0.442 |
| Default | all submissions | all submissions | sum | 3.792 | 0.455 |
| Default | reference, all submissions | all submissions | sum | 3.564 | 0.442 |

We then ran a MQM human evaluation to collect the ground-truth annotations for these same transla­tions and report the character-level F1 meta-evaluation metric (Blain et al., 2023). In comparison to ground-truth annotations from the human MQM evaluation, ICL with Gemini achieves a character-level F1 meta-evaluation metric (Blain et al., 2023) of only 19.14. These results highlight the loss in annotation accuracy incurred when utilizing model-based annotation of online data (required for reinforcement learning approaches). See 8 for the performance of our fine-grained annotator model on the WMT'20-'21 test sets.

Table 8: Character-level F1, precision, and recall of our fine-grained annotator model when anno­tating outputs from our base translation model, computed with respect to human MQM annotations collected for the same translations.

| Character-Level F1 | Precision | Recall |
|--------------------|-----------|--------|
| 19.14 | 14.34 | 28.76 |

# D  SAMPLE TRANSLATIONS

Below, we present candidate translations from the Zh→En experiment. Examples were chosen to emphasize differences between methods while representing a diversity of translation lengths. Concretely, we subsetted to examples where the edit distance between any pair of methods was at least 15. Then, we stratified examples into equal-sized bins based on the length of the TWA translation and chose one example from each bin. Within each strata, the example was chosen qualitatively based on ease of understanding and diversity in content.

| Reference | So I simply waited patiently, when I checked my phone at midday, the order was still in the status and had not been delivered, I immediately contacted the customer service and requested that the order be canceled as quickly as possible, customer service responded that the operation was well. |
|-----------|---|
| TWA | Well, I waited patiently, and when I looked at the phone at noon, the order was still in the state of delivery, so I immediately contacted the customer service to request that the order be cancelled as soon as possible, and the customer service responded that it was OK. |
| SFT | Well, I waited patiently, and when I looked at the phone at noon, the order was still in the state of delivery, and immediately contacted the customer service to ask for the order to be cancelled quickly, and the customer service responded that it was OK. |
| DPO | I was nervous about the timing of the delivery, but when I saw the order was still in the order and the order had not been shipped, I immediately contacted the customer service to request that the order be cancelled. |

| Reference | 5 Users may substitute more readily available substances, such as alcohol or benzodiazepines, or combine opioids with synthetic narcotics as a result of opioid shortages brought on the COVID-19 regulations. |
|---|---|
| TWA | The shortage of opioids caused by the new coronavirus restrictions may cause users to replace them with more readily available substances such as alcohol or benzodiazepines, or mix opioids with synthetic drugs. |
| SFT | The shortage of opioids caused by the new coronary pneumonia restriction measures may lead users to replace them with more readily available substances such as alcohol or benzodiazepines, or mix opioids with synthetic drugs. |
| DPO | Opioid shortages caused by new coronavirus restrictions may lead users to be replaced with more readily available substances such as alcohol or benzodiazepines or to mix opioids with synthetic drugs. |

| Reference | Ear covers come with the earphones, use earbuds and bottom text if uncomfortable. |
|---|---|
| TWA | The headphones themselves are equipped with medium-sized ear sails, if you feel discomfort to ask the text to blend in small though the outside. |
| SFT | The headphones themselves are equipped with medium-sized ear sails, if they feel unwell to ask the text to merge small though outside the machine. |
| DPO | The headphones themselves are fitted with medium-sized sails, which can be used if the wearer feels unwell and needs to be externally fitted. |

| Reference | It makes no difference if the seller's company has already collected the money and does not follow up. |
|---|---|
| TWA | It doesn't matter if the seller company has also received money anyway and doesn't follow up. |
| SFT | It doesn't matter if the seller company has also received money and doesn't follow up anymore. |
| DPO | It doesn't matter if the seller has taken the money or not followed up. |

| Reference | To turn on, press and hold for roughly 1 second |
|---|---|
| TWA | Press long about 1 second to start the machine |
| SFT | Start up in about 1 second |
| DPO | Press on for about 1 seconds to boot |

