# OpenReview forum: "Learning from others' mistakes: Finetuning machine translation models with span-level error annotations"
_ICLR.cc/2025/Conference — Submitted to ICLR 2025_

### Official Review · Reviewer_FV2i · 2024-11-02

**Soundness:** 2
**Presentation:** 3
**Contribution:** 2
**Rating:** 5
**Confidence:** 5

**Summary:**

This paper proposes a fine-grain loss to penalty error during supervised fine-tuning (SFT) for machine translation. The main contribution is to take the annotation of the MQM dataset (fine-grained, human-label translation errors) as both positive and negative supervision during SFT at the token level. The main results of this paper are compared with those of using DPO and SFT in two language directions, EN-DE and ZH-EN, showing some improvements in their setting. Also, ablation studies clearly show the difference among multiple variants of their methods.

The writing is clear and easy to follow. The method, to some extent, might inspire some developments in nowadays optimization toward human preference. However, I hold some concerns with their motivation and evaluation, see the weakness part.

**Strengths:**

1. The writing is clear and easy to follow.
2. MQM shows fine-grained error annotation. Exploring and leveraging MQM data for MT training is interesting. Also, it might inspire some research in optimizing translation towards human preferences.
3. Positive results in two language directions under their settings.

**Weaknesses:**

1. MQM data is hard to largely achieve: Compared to other MT-eval annotation data at the sentence level, like DA, MQM data shows more detailed human evaluation. However, it is also hard to largely achieve (even for the DA dataset, it only covers 10+ languages and hundreds k samples through years of work done by WMT).

2. Feasibility aside, if we only focus on the generality of this technique, this method is hard to generalize to other domains, like QA, as it is hard to say that span annotation also applies to QA data collection.

3. The baseline is not strong: 1) The baseline model leg behind the average performance of WMT submission quite a lot. 2) In Table 3, the SFT setting improves results a lot. This gain from SFT is weird if their base model is strong. It would be much better if they could simply increase the model size and clean data for base model training.

Suggestions:
1. Since DPO and SFT are concepts from the LLM community, it would be beneficial to show results on LLM-based MT. (I don't believe it's essential.)

**Questions:**

1. In Table 3, the main baseline I wanna see is applying SFT on reference data, while it is missing. I am speculating that the gain of this method is mainly from removing noise (which might even exist in the filtered submission data) based on human annotation. If so, the mechanism of success is far away from guiding translation based on negative signals. To resolve this concern, could you show some results of SFT on reference?

2. As mentioned in Weakness-3, could you show some results of Table-3 when using a stronger model? E.g., apply TWA on M2M100 or NLLB.

3. In lines 201-202, does it simply indicate truncating loss to the first error token? If so, some loss truncation methods could be compared, like https://arxiv.org/pdf/2407.02208

4. In Table 1 and Table 3, the scores for the base model are not aligned? Could you explain a little bit about my misunderstanding?

---

> ### Author Response · Authors · 2024-11-27
> **Author Response to Reviewer FV2i (part 1)**
>
> Thank you for reviewing our work. Responding to your individual points:
> 1. [MQM data is hard to largely achieve: Compared to other MT-eval annotation data at the sentence level, like DA, MQM data shows more detailed human evaluation. However, it is also hard to largely achieve (even for the DA dataset, it only covers 10+ languages and hundreds k samples through years of work done by WMT).]
> - We agree with the reviewer’s description of the trade-off between information density and annotation expense with MQM vs. DA. However, even though MQM data may be expensive to collect, we believe that there is immense potential even in just effectively utilizing existing data, as well as data that will be collected anyway for the evaluation of MT systems.
> 2. [Feasibility aside, if we only focus on the generality of this technique, this method is hard to generalize to other domains, like QA, as it is hard to say that span annotation also applies to QA data collection.]
> - We respectfully disagree. In fact, Wu et al 2023 [[1]](https://arxiv.org/pdf/2306.01693) show in a human study on long-form QA that annotators spend a comparable amount of time providing fine-grained feedback as they do providing preference feedback. Given the similar amounts of effort, QA is likely a promising domain to consider annotating fine-grained feedback for. Other such settings with promise could include reducing toxicity or hallucinations, as annotating a sequence as toxic or “with hallucination” requires identifying the instance of toxicity or hallucination as a precursor.
> 3. [The baseline is not strong: 1) The baseline model lagged behind the average performance of WMT submission quite a lot. 2) In Table 3, the SFT setting improves results a lot. This gain from SFT is weird if their base model is strong. / Could you show some results of Table-3 when using a stronger model? E.g., apply TWA on M2M100 or NLLB.]
> - We agree that to achieve state-of-the-art performance, we would want to invest more effort in building a stronger pretrained base model. In our case, we used only the WMT’23 training data in our experiments to tightly control all the data seen by the model. Even so, the fact that TWA is able to improve performance on top of the gains from SFT demonstrates the promise of the proposed method in utilizing annotation information, not just exploiting a different translation quality between model and data.
> - We were unfortunately unable to rerun the experiments from Table 3 with a different model, but we were able to run TWA using MQM annotations of the model’s own translations, in order to simulate the setting where the base model is of comparable quality to the data used in TWA (new section 6.5). We find in this setting that TWA significantly improves performance, from 4.203/0.429 Metric-X/COMET to 3.710/0.456 Metric-X/COMET. This suggests that TWA can offer performance improvements based on annotation information alone, even when there is no difference in translation quality between the model and finetuning data.
> 4. [Since DPO and SFT are concepts from the LLM community, it would be beneficial to show results on LLM-based MT. (I don't believe it's essential.)]
> - Thanks for the suggestion. We were not able to run these experiments at this time, but we agree with the reviewer that it would be useful future work!
>
> (continued in part 2)

---

> > ### Author Response · Authors · 2024-11-27
> > **Response to Reviewer FV2i (part 2)**
> >
> > (continued from part 1)
> >
> > 5. [In Table 3, the main baseline I wanna see is applying SFT on reference data, while it is missing. I am speculating that the gain of this method is mainly from removing noise (which might even exist in the filtered submission data) based on human annotation. If so, the mechanism of success is far away from guiding translation based on negative signals. To resolve this concern, could you show some results of SFT on reference?]
> > - Thanks for the question. To the reviewer’s question about whether the method is “guiding translation based on negative signals” or simply “removing noise…based on human annotation,” we believe the clearest demonstration of the distinction can be found in the ablations (Table 4). There, we see that explicitly utilizing a negative loss on error tokens outperforms simply ignoring those tokens in the loss, suggesting that the explicit negative signal is indeed helpful.
> > - As for comparing with SFT on references alone, please see the table below (En->De on left, Zh->En on right). SFT on references alone tends to perform better on Metric-X while TWA performs better on COMET, though the differences are small in all cases. Both significantly outperform SFT on references and filtered submissions however, even though SFT on references uses less data than SFT + filter while TWA uses more data. This suggests different mechanisms are likely at play: SFT on references outperforming SFT + filter suggests that even the error-free submissions may not be as high-quality as the references in this dataset; TWA outperforming SFT + filter suggests that considering all annotated submissions (including imperfect ones) intelligently can outperform ignoring negative information altogether.
> >
> > | Model | Submissions | References | Metric-X (↓) | COMET (↑) | Metric-X (↓) | COMET (↑) |
> > |---|---|---|---|---|---|---|
> > | SFT |  | ✓ | **2.851** | 0.511 | **3.960** | 0.278 |
> > | TWA | ✓ | ✓ | 2.882 | **0.513** | 3.965 | **0.290** |
> > | SFT + filter | ✓ | ✓ | 2.950 | 0.499 | 4.004 | 0.289 |
> >
> > 6. [In lines 201-202, does it simply indicate truncating loss to the first error token? If so, some loss truncation methods could be compared, like https://arxiv.org/pdf/2407.02208]
> > - Lines 201-202 indicate truncating the loss on token after the end of the first error span. We agree that it could be interesting to test other methods, but methods such as the one linked rely on the accuracy of the original model’s distribution as an indication of the correctness of an example sequence, which may not be a good assumption in the starting model setup of this paper.
> > 7. [In Table 1 and Table 3, the scores for the base model are not aligned? Could you explain a little bit about my misunderstanding?]
> > - Great question. Table 1 denotes the scores of the base model translations of sources in the training set (i.e., WMT’20 and ‘21 MQM data), to directly compare to the submissions data being used to finetune the MT model. Table 3, on the other hand, looks at the eval set (WMT’23).
> >
> > We would like to thank you once again for your thoughtful review. We hope that our response has sufficiently addressed your concerns and that you may be willing to reconsider your score.

---

### Official Review · Reviewer_hXfV · 2024-11-03

**Soundness:** 2
**Presentation:** 3
**Contribution:** 2
**Rating:** 3
**Confidence:** 3

**Summary:**

The authors propose to train machine learning systems with error annotations, where both the reference translation and a poor translation are given. Here, the poor translation is annotated by humans, indicating which spans of the text are wrong (and how wrong it is). The authors propose to use an unlikelihood loss to discourage the model to generate tokens in the error spans.

**Strengths:**

1. The authors utilize existing data annotations and show it’s helpful to train machine learning systems.
2. The authors compare their method with DPO.

**Weaknesses:**

1. Novelty. The main novelty of this work is utilizing additional annotations to improve translation systems, which is not surprising. Otherwise, the proposed unlikelihood training is straightforward.
2. Getting annotations is costly. The authors propose to utilize existing annotations, which is scarce. Although in the limited data setting, the proposed method is better than DPO, it’s likely that DPO is still much better in terms of annotation cost.
3. Relatively weak experimentation. The authors only evaluated two translation directions in one dataset, which may be below standard practice of translation papers.

**Questions:**

How efficient is TWA training compared with SFT?

---

> ### Author Response · Authors · 2024-11-27
> **Author Response to Reviewer hXfV**
>
> Thank you for your review. Addressing your concerns:
>
> 1. [Novelty. The main novelty of this work is utilizing additional annotations to improve translation systems, which is not surprising. Otherwise, the proposed unlikelihood training is straightforward.]
> - In terms of novelty:
>   - This work is the first to propose utilizing MQM data, a readily available source of data in the MT community, to directly finetune MT models. The setting of training on offline annotated data is also a relatively underexplored area.
>   - The proposed method is the first to combine the specific concepts of unlikelihood on errors, span-level error loss, and ignoring off-trajectory tokens to utilize offline, annotated data.
> 2. [Getting annotations is costly. The authors propose to utilize existing annotations, which is scarce. Although in the limited data setting, the proposed method is better than DPO, it’s likely that DPO is still much better in terms of annotation cost.]
> - It’s worth noting that in this setup, annotating preference pairs directly would be expensive as we find it to be important to consider every possible pair, i.e., (10 choose 2) = 45 pairs for every source. This is in contrast to annotating each of the ~10 translations per source separately. One could alternatively annotate the translations at a sequence level rather than a finer-grained level and then construct preference pairs programmatically. However, in the case of assigning MQM scores, a sequence-level MQM score is the sum of the scores of the error spans, making the fine-grained annotation of TWA comparable to the sequence-level annotation needed to construct data for DPO.
> 3. [Relatively weak experimentation. The authors only evaluated two translation directions in one dataset, which may be below standard practice of translation papers.]
> - Respectfully, we disagree that this is below standard practice; for instance, multiple translation papers accepted at ICLR last year experimented with two language pairs, e.g., [[1]](https://openreview.net/pdf?id=3KDbIWT26J)[[2]](https://openreview.net/pdf?id=bkNx3O0sND)[[3]](https://openreview.net/pdf?id=XTHfNGI3zT). We do, however, agree with the reviewer that more language pairs would be better and are working to add an additional language pair (results forthcoming).
> 4. [How efficient is TWA training compared with SFT?]
> - TWA training is as efficient as SFT, utilizing only one model in memory (in contrast to DPO, which requires two models) and needing just a single forward pass to compute the loss.
>
>
> Thanks again for your review. Please let us know if you have any additional questions or concerns. If not, we would be grateful if you would reconsider your score.

---

> ### Comment · Reviewer_hXfV · 2024-12-01
>
> Thanks for the response! A few quick points:
>
> > 1. The proposed method is the first to combine the specific concepts.
>
> I agree with the authors that the combination may be novel, but I also think the concepts individually aren't very novel. I am happy to leave the decision to the chairs.
>
> > 2. annotating preference pairs directly would be expensive as we find it to be important to consider every possible pair
>
> I may not understand the authors' point fully. I assumed if it is important to consider every possible pair, that would also apply to annotating error spans. All in all, I appreciate the discussion, which perhaps can be checked with concrete experiments.
>
> Overall, none of my concerns are really addressed, and I maintain my initial assessment.

---

> > ### Author Response · Authors · 2024-12-04
> > **Author Response to Reviewer hXfV**
> >
> > We thank the reviewer for their feedback and would like to address the remaining concerns.
> >
> > Regarding the additional results:
> >
> > As mentioned in our response to Reviewer Uwxh, we have added results for a new evaluation metric (Bleurt) and an additional language pair (en→zh). We hope these new experiments demonstrate the robustness of our approach and address the reviewer’s concerns about the breadth of experimentation.
> >
> > On the cost comparison between DPO and TWA:
> >
> > 1. Annotation effort: It is not obvious that span-level annotations for TWA are inherently more expensive than preference labels for DPO, since both tasks can be demanding (i.e. DPO requires careful comparison of both translations).
> >
> > 2. Scaling: Annotating preference pairs for DPO involves considering every possible pair (as we find it to be important to consider every possible pair in our experiments), scaling as O(n^2), whereas TWA annotations scale linearly with O(n). This difference significantly impacts cost as the number of translations increases.
> >
> > 3. Quality vs. cost: While cost is an important factor, we believe quality should be the primary criterion when evaluating methods for improving machine translation.

---

### Official Review · Reviewer_uwxh · 2024-11-03

**Soundness:** 3
**Presentation:** 3
**Contribution:** 2
**Rating:** 5
**Confidence:** 3

**Summary:**

This work investigates improving machine translation performance through fine-grained, span-level human-crafted annotations. It introduces a hybrid training loss that treats error spans as negative partial samples, ignores tokens after the first error, and considers tokens before the first error as positive samples. This fine-tuning approach is termed TWA. TWA is then compared against a set of newly proposed baselines, demonstrating outstanding performance.

**Strengths:**

* The paper is easy to follow.
* It motivates exploration of learning from detailed, fine-grained signals.
* The discussion on the importance of allowing the model to learn which tokens in an error span should be penalized is clear and well-motivated. The experiment supporting this claim is appropriately designed.

**Weaknesses:**

* Training Data Overlap: MetricX-23 is fine-tuned on MQM WMT’20-’21, and TWA is also trained on this dataset. This overlap suggests that evaluation might leak into training, disqualifying MetricX-23 as an evaluation metric in this setup.
* Motivation for Task: The task is not well-motivated. Obtaining fine-grained annotations is costly, and it’s unclear why methods are needed to utilize this type of supervision. Although this is discussed in the third paragraph of the conclusion, it comes too late (it is better to be addressed in the Introduction), and the motivation largely pertains to other tasks that might benefit from TWA techniques. This raises the question: why focus on machine translation instead of these other tasks?
* Choice of Offline Learning: It’s not well-explained why offline learning is favored over RL-based models. Efficiency might be one reason, which could benefit from further discussion and experimental analysis.
* Design Choice Clarity: The design choice mentioned in footnote 1 on page 4 lacks adequate explanation.
* Evaluation Choices: The choices of evaluation metrics and experimental designs are not well-justified.
* Statistical Analysis in Section 6.1: The statistical test mentioned in Section 6.1 lacks detail. It’s unclear what test is used or how it’s conducted. More clarity here would improve the reader's understanding, especially when there is only one instance of each model.
* Baseline Selection: The baselines are loosely defined. While there are efforts to select the best variant of DPO, the approaches cited as baselines remain relatively simple and open to criticism. For example, why not consider weighted sampling instead of TWA-seq, or use erroneous samples as negative samples instead of Filter + SFT? Similarly, why not adopt a weighted contrastive learning approach rather than DPO? Additionally, it raises questions as to why RL-based methods are excluded as baselines. Moreover, for baselines that do not require fine-grained supervision, other larger and less costly datasets could be leveraged. Restricting models with fewer training data limitations to the same dataset may be unfair.
* Impact of Ignoring Off-Trajectory Tokens: The observation that ignoring off-trajectory tokens benefits one translation path while impairing another needs further exploration, even though it’s noted as a topic for future work. Given that ignoring these tokens is presented as a critical step—likely essential for En->De to outperform baselines—it would be beneficial to discuss this more thoroughly. Experiments across more translation paths might shed light on this factor’s impact. Additional analysis to identify the underlying reasons is necessary.
* Further Elaboration on Observation in Sec. 6.3: Observation mentioned in Sec. 6.3 would benefit from additional elaboration.
* Experiment in Figure 2: The experiment illustrated in Figure 2 highlights the importance of allowing the model to learn which tokens within an error span should be penalized. While the presentation is intuitive, including more statistical evidence and quantitative analysis would strengthen this point.
* Expansion of Translation Paths and Metrics: It’s suggested to test additional translation paths and incorporate more evaluation metrics, as the two currently provided are not strongly correlated.
* Marginal Performance Gap with References: In the setup that utilizes References, the performance gap between TWA and other baselines is minimal. A stability test could help substantiate the claims more effectively.
* Minor Weaknesses:
    * Line 064: “Training with Annotations (TWA)” is repeated (the abbreviation alone suffices) and is incorrectly linked.
    * Lines 124-126: Missing a verb, rendering the sentence incomplete.
    * Unaddressed Observations on TWA: TWA’s performance lagging behind DPO in one experiment is not addressed in the analysis.

**Questions:**

I would appreciate clarification on the questions raised in the weaknesses section. Additionally, please let me know if there are any other aspects I may have overlooked that could address areas of confusion.

---

> ### Author Response · Authors · 2024-11-27
> **Author Response to Reviewer uwxh (part 1)**
>
> Thank you for your thorough review. Addressing each comment under weaknesses point-by-point:
> 1. [Training Data Overlap: MetricX-23 is fine-tuned on MQM WMT’20-’21, and TWA is also trained on this dataset. This overlap suggests that evaluation might leak into training, disqualifying MetricX-23 as an evaluation metric in this setup.]
> - As mentioned in section 5.4, we intentionally use MetricX-23, which is fine-tuned on MQM WMT’20-’21, alongside COMET-20, which is not, in order to test the benefits of the proposed approach under an evaluation that is sensitive to the specific information found in MQM data as well as one that is not. This guards against over-indexing to either evaluation.
> 2. [Motivation for Task: The task is not well-motivated. Obtaining fine-grained annotations is costly, and it’s unclear why methods are needed to utilize this type of supervision. Although this is discussed in the third paragraph of the conclusion, it comes too late (it is better to be addressed in the Introduction), and the motivation largely pertains to other tasks that might benefit from TWA techniques. This raises the question: why focus on machine translation instead of these other tasks?]
> - The reason we focus on machine translation is because it is an application where large amounts of fine-grained information is already readily available. Positive results in this setting can motivate the collection of fine-grained results in other settings, which need not be more expensive than sequence-level annotations in many situations. We have incorporated additional discussion in the introduction as suggested; thank you for the suggestion.
> 3. [Choice of Offline Learning: It’s not well-explained why offline learning is favored over RL-based models. Efficiency might be one reason, which could benefit from further discussion and experimental analysis.]
> - As mentioned in the related work, we do not directly benchmark against RL-based methods as they 1. are more memory intensive, requiring at least one additional model in memory to output rewards, 2. are much more difficult to optimize than direct finetuning methods, and 3. do not take advantage of the (already available) offline examples themselves, besides learning a reward model to predict their annotations.
> 4. [Design Choice Clarity: The design choice mentioned in footnote 1 on page 4 lacks adequate explanation.]
> - Thanks for the feedback. We have updated it to: “Under the MQM rating system, some major errors are given a score of -25 (namely those categorized as non-translations), but we use a weight of -5 for these errors as well.” The primary reason was for simplicity.
> 5. [Evaluation Choices: The choices of evaluation metrics and experimental designs are not well-justified.]
> - See above on our motivation for including Metric-X and why we do not compare against RL methods.
> 6. [Statistical Analysis in Section 6.1: The statistical test mentioned in Section 6.1 lacks detail. It’s unclear what test is used or how it’s conducted. More clarity here would improve the reader's understanding, especially when there is only one instance of each model.]
> - We’ve updated the section describing the test in section 6.1. In short, each model is associated with a distribution over source-translation scores (Metric-X or COMET), and we run a permutation test between all pairs to see if results are statistically significant under the null that the scores for each system come from the same distribution. Then, we turn these pairwise significance results into a global ranking via a greedy algorithm that creates a new cluster when a new system is significantly worse than any of the prior systems.
>
> (continued in part 2)

---

> > ### Author Response · Authors · 2024-11-27
> > **Response to Reviewer uwxh (part 2)**
> >
> > (continued from part 1)
> >
> > 7. [Baseline Selection: The baselines are loosely defined. While there are efforts to select the best variant of DPO, the approaches cited as baselines remain relatively simple and open to criticism. For example, why not consider weighted sampling instead of TWA-seq, or use erroneous samples as negative samples instead of Filter + SFT? Similarly, why not adopt a weighted contrastive learning approach rather than DPO? Additionally, it raises questions as to why RL-based methods are excluded as baselines. Moreover, for baselines that do not require fine-grained supervision, other larger and less costly datasets could be leveraged. Restricting models with fewer training data limitations to the same dataset may be unfair.]
> > - We address each alternative baseline proposed by the reviewer below:
> >   - Weighted sampling: Filter + SFT can be considered a version of weighted sampling, i.e., only using non-erroneous samples
> >   - Using erroneous samples as negative samples: TWA-seq does this
> >   - Weighted contrastive learning approach: If the reviewer has a particular reference or method instantiation in mind (e.g., where to put weights, how to choose weights, etc.), it could be interesting to consider, but ultimately the reason we use DPO is that it is an established method in the field and thus likely a method one might think to turn to.
> >   - RL: see above for why we do not test RL methods.
> >   - More data with baselines that do not require fine-grained supervision: the goal of this work is to develop a method to be able to better take advantage of fine-grained information than existing methods. This is why we compare methods while controlling the dataset.
> > 8. [Impact of Ignoring Off-Trajectory Tokens: The observation that ignoring off-trajectory tokens benefits one translation path while impairing another needs further exploration, even though it’s noted as a topic for future work. Given that ignoring these tokens is presented as a critical step—likely essential for En->De to outperform baselines—it would be beneficial to discuss this more thoroughly. Experiments across more translation paths might shed light on this factor’s impact. Additional analysis to identify the underlying reasons is necessary.]
> > - Thanks for the question. It’s worth noting that the version of TWA that does not ignore off-trajectory tokens still outperforms baselines on En->De (3.325 MetricX and 0.495 COMET versus 3.573 and 0.481 for the next best baseline). As for understanding why ignoring off-trajectory tokens significantly improves performance for En->De while incurring no benefit for Zh->En, we were not able to run sufficient experiments to pinpoint the root cause, but one potential reason could be the fact that the Zh->En submissions are generally lower quality to begin with, meaning there is less room to differentially improve performance with fine-grained information (as evidenced by the fact that TWA is better than TWA-seq but not significantly so). We agree that future experiments along more translation paths could help elucidate the reasons for differences, but given the significant improvement offered to one code path and minimal effect to another, we believe the strategy to be a useful one to consider.
> > 9. [Further Elaboration on Observation in Sec. 6.3: Observation mentioned in Sec. 6.3 would benefit from additional elaboration.]
> > - We added additional discussion to the section which contrasts the negative loss candidates based on their contribution to the loss and gradient as the span moves towards its desired result.
> > 10. [Experiment in Figure 2: The experiment illustrated in Figure 2 highlights the importance of allowing the model to learn which tokens within an error span should be penalized. While the presentation is intuitive, including more statistical evidence and quantitative analysis would strengthen this point.]
> > - Thanks for the suggestion. We have added the following experimental result to the section: Quantitatively, using a token-level unlikelihood loss on En->De submissions for every error token achieves a Metric-X of 3.433 and COMET of 0.470, whereas using a span-level loss achieves a Metric-X of 3.325 and COMET of 0.495.
> > 11. [Expansion of Translation Paths and Metrics: It’s suggested to test additional translation paths and incorporate more evaluation metrics, as the two currently provided are not strongly correlated.]
> > - For evaluation metrics, we specifically chose metrics that were distinct in the data they were trained on for a more holistic evaluation, and while the rank order of some baselines is different between the evaluation metrics, TWA is consistently rank 1 under both metrics. We are working on incorporating experiments for an additional language pair now.
> >
> > (continued in part 3)

---

> > > ### Author Response · Authors · 2024-11-27
> > > **Response to Reviewer uwxh (part 3)**
> > >
> > > (continued from part 2)
> > >
> > > 12. [Marginal Performance Gap with References: In the setup that utilizes References, the performance gap between TWA and other baselines is minimal. A stability test could help substantiate the claims more effectively.]
> > > - We agree with the reviewer that the differences across methods become smaller when references are included, but TWA is still significantly better than baselines on En->De and in the rank-1 cluster for Zh->En. These results are based on pairwise permutation tests of significance.
> > > 13. [Minor Weaknesses: Line 64: “Training with Annotations (TWA)” is repeated. Lines 124-126: Missing a verb, rendering the sentence incomplete.]
> > > 	Fixed, thanks!
> > > 14. [TWA’s performance lagging behind DPO in one experiment is not addressed in the analysis.]
> > > - TWA is never significantly worse than DPO in the experiments. In the Zh->En result where DPO looks better in Metric-X (though not significantly so), it is substantially worse on COMET (less than half the COMET score of TWA), suggesting that DPO has exploited an idiosyncrasy of the Metric-X model without truly improving in overall performance. We have added this point to the results discussion.
> > >
> > > Thank you again for your review. Please let us know if you have any remaining questions or concerns; otherwise, we would greatly appreciate it if you would consider raising your score.

---

> > > > ### Comment · Reviewer_uwxh · 2024-11-28
> > > >
> > > > Thank you for your efforts and response.
> > > >
> > > > 12. I believe including more translation pairs addresses this concern as well. If TWA is always in the first cluster, while other approaches come and go, then it is possible to claim the stability of the superiority of TWA. I would love to see the results on other translation pairs.
> > > >
> > > > 13. Thank you for for the explanation. I believe my suggestion in the 1st point in our discussion series (to include another source-text-ignorant metric) can well support your claim here (that DPO has exploited an idiosyncrasy of the Metric-X model), proving the promise of TWA, ineffectiveness of DPO, and the importance of reporting Metric-X (as an overfit-control metric). It not only proves the superiority of the TWA, but also provides insights about the Metric-X (as the leaked metric) being untrustworthy, which is interesting to see.
> > > >
> > > >
> > > > Thanks again for your efforts to address all concerns. I am considering and would be happy to increase my score. However, I am still concerned about the unresolved points (1, 11, 13 concerning evaluation metric; 8, 11, 12 concerning language pairs; and 3, 4 as remaining questions/discussions).

---

> > > > > ### Author Response · Authors · 2024-12-03
> > > > > **Response to Reviewer uwxh**
> > > > >
> > > > > Thanks for your response! We are glad to have addressed many of your concerns in our previous response. As for your remaining concerns:
> > > > >
> > > > > 1 / 11 / 13) [I suggest including also another source-text-ignorant metric, which is not based on the same training data.]
> > > > >
> > > > > Great idea. We’ve included BLEURT as an additional evaluation metric, which is source-text-ignorant and has not been trained over the MQM data. While we can no longer update the paper, please see the updated table below
> > > > >
> > > > > | Model | Submissions | References | Metric-X (↓) | COMET (↑) | Bleurt (↑) |Metric-X (↓) | COMET (↑) |Bleurt (↑) |
> > > > > |---|---|---|---|---|---|---|--|--|
> > > > > |SFT | ✓ | | 3.573 | 0.481 | 0.658 | 4.253 | 0.255 | 0.650 |
> > > > > |DPO| ✓ | | 3.792 | 0.455 | 0.664 | **4.072** | 0.113 | 0.615 |
> > > > > |TWA| ✓ | | **2.944** | **0.507** | **0.668** | 4.091 | **0.277** | **0.651** |
> > > > > |SFT | ✓ | ✓ | 3.159 | 0.491 |  0.662 | 4.094 | 0.271 | 0.652  |
> > > > > |DPO| ✓ | ✓ | 3.564 | 0.442 |  0.660 | 4.063 | 0.113 |  0.614 |
> > > > > |SFT + filter | ✓ | ✓ | 2.950 | 0.499 | 0.670  | 4.004 | 0.289 |  0.652 |
> > > > > |TWA-seq | ✓ | ✓ | 3.158 | 0.485 | 0.663  | 3.993 | 0.284 | 0.652 |
> > > > > |TWA| ✓ | ✓ | **2.882** | **0.513** | **0.672** | **3.965** | **0.290** | **0.653**  |
> > > > >
> > > > > 8 / 11 / 12) [Another language pair]
> > > > > We will post results for an additional language pair as a separate response.
> > > > >
> > > > > 3) [Comparing to RL. -1- This claim requires supporting experiments. You can have a control experiment, fixing the memory cost of both approaches, and compare the results. -2- Fair. FYI, the caption of Figure 3.b is the same as that of 3.a, which is probably a copy-paste mistake (it has 10^4 value as proportion) -3- It is solvable by offline RL; one may develop an RL-based agent, once trained using offline RL on the offline examples, then start learning from online samples.]
> > > > >
> > > > > -1- An RL-based approach would require at least two models, the current model being trained as well as a learned reward model; plus, to avoid reward over-optimization, it is standard to additionally add some sort of regularization with respect to the original model, leading to a model count of 3. TWA, in contrast, requires keeping only one model in memory. Thus, controlling for memory would require using a much smaller starting model and thus a worse starting point for RL.
> > > > >
> > > > > -2- Thanks for catching the typo–we’ve corrected the caption to “Number of error tokens in each output sequence.”
> > > > >
> > > > > -3- From [1]: “With virtually no tuning of hyperparameters, DPO performs similarly or better than existing RLHF algorithms, including those based on PPO.” Also, RL-based methods suffer from the overoptimization problem in machine translation as shown in [2]: “Our preliminary experiment observed that as the reward increases, the translation performance deteriorates. This phenomenon is dubbed as overoptimization.”. Hence, we focused our efforts on DPO as a strong baseline for preference optimization.
> > > > >
> > > > > [1] “Direct Preference Optimization: Your Language Model is Secretly a Reward Model”, Neurips 2023, by Rafailov et al.
> > > > >
> > > > > [2] “Improving Machine Translation with Human Feedback: An Exploration of Quality Estimation as a Reward Model”, ACL 2024, by Zhiwei He et al.
> > > > >
> > > > > 4) [Motivation for -5 instead of -25]
> > > > >
> > > > > We agree that implementation-wise, adding an additional reward value is trivial. We consider this as an additional hyperparameter of TWA; further tuning of this hyperparameter could improve the performance of TWA. Our experiments in the draft show that even without this hyperparameter tuning, TWA outperforms all the baselines. We decided a priori to avoid the large magnitude reward of -25 due to possible negative effects on optimization (though not thoroughly confirmed empirically).

---

> > > > > > ### Comment · Reviewer_uwxh · 2024-12-03
> > > > > >
> > > > > > Thank you for your efforts.
> > > > > >
> > > > > > I believe Bleurt (↑) significantly improves the soundness of the experiment and supports your claims effectively. As can be seen, while TWA significantly improves COMET in both setups, it either improves or, at the very least, does not harm Bleurt, which is an undeniable metric. Moreover, the results strongly support your claim that DPO exploits an idiosyncrasy of the Metric-X model without truly improving overall performance, as it actually harms Bleurt, whereas TWA, despite performing worse on Metric-X, results in a better Bleurt score. I strongly suggest including these results and adding appropriate discussion, specifically regarding DPO’s exploitation of Metric-X, in the final revision.
> > > > > >
> > > > > > 3-1- The number of the loaded models does not necessarily imply the cost.
> > > > > >
> > > > > > 3-3- I would suggest to include the citations in the final revision.
> > > > > >
> > > > > > 4. While your intuition is sound in my opinion, there is a lack of experimentation, which is understandable considering the limited time and space. Thank you for clarification. I believe it can be a (minor) future exploration.
> > > > > >
> > > > > > I am looking forward to see the impact of ignoring off-trajectory tokens in the additional language pair.

---

> > > ### Comment · Reviewer_uwxh · 2024-11-28
> > >
> > > Thank you for your efforts and response.
> > >
> > > 7. About the first two bullets, a variant can have both information. About the last bullet, I don't think it can be seen as a controlled experiment to compare models limiting to a dataset friendly to only one of them. Ideally, we should control the data preparation cost (each data entry with fine-grained annotation would cost more than one entry with sentence-level annotation, thus we can have more of the latter). However, it is understandable that such a controlling is not perfectly possible. Yet, if one model has to go into disadvantage when comparing, it is always better to be the model that is claimed to be superior. Seeing the already-included results of models not incorporating fine-grained annotation is interesting, but does not tell much about the comparison. Explanation of the other bullets are fair. Thank you.
> > >
> > > 8. Thank you for the clarification. By "essential to outperform baselines," I am mainly referring to Filter+SFT. It is understandable that not all the observations get justified in one single work. Yet, it is possible to shed light on this matter by including more translation directions.
> > >
> > > 9. Thank you. The newly added explanation sounds reasonable to me.
> > >
> > > 10. Thank you so much. The included statistics are awesome.
> > >
> > > 11. For metrics, please refer to my response to the 1st point in our discussion series. For language pairs, thank you for your efforts. I believe it would improve the quality of your work substantially (as I also mentioned in the 8th point above).

---

> > ### Comment · Reviewer_uwxh · 2024-11-28
> >
> > Thank you for your efforts and response.
> >
> > 1. Given your explanation, MetricX-23 is representing as a MQM-sensitive metric. However, you introduce it as the representative of source-text-ignorant metrics:
> >
> > > MetricX-23 is a reference-based metric which scores a translation based on a reference and a hypothesis, without taking into account the source text. COMET-20 takes into account the source text, hypothesis, and reference translation.
> >
> > I agree that your explanation justifies your choice to include MetricX-23, and it shows your great critical thinking. However, this metric is not qualified to also represent source-text-ignorant metrics. I suggest including also another source-text-ignorant metric, which is not based on the same training data.
> >
> > 2. Perfect. I would love to also see such an honest clarification of your motivation to choose MT somewhere in the paper (e.g., in the third paragraph of Discussion, the third paragraph of Introduction, or a footnote). Moreover, there is a missing citation in L40.
> >
> > 3. -1- This claim requires supporting experiments. You can have a control experiment, fixing the memory cost of both approaches, and compare the results. -2- Fair. FYI, the caption of Figure 3.b is the same as that of 3.a, which is probably a copy-paste mistake (it has 10^4 value as proportion) -3- It is solvable by offline RL; one may develop an RL-based agent, once trained using offline RL on the offline examples, then start learning from online samples.
> >
> > 4. Sorry about the confusion. By lacking clarity I meant lack of motivation and purpose of choice. I can see you mention it is due to simplicity, but I cannot see how it simplifies the approach. It shouldn't be a big deal to add one possible reward value. Please let me know if I am missing something.
> >
> > 5. Fair.
> >
> > 6. Interesting. Thank you for the clarification.

---

> ### Comment · Reviewer_uwxh · 2024-12-03
>
> Dear Authors,
>
> I hope this message finds you well.
>
> Given the valuable discussions we've had, I am happy to increase my score. Unfortunately, due to my timezone, I am unable to wait until the last minute for your final responses to the remaining points. I would be happy to leave the interpretation of any further responses to the chairs.
>
> Thank you for your commitment to the discussions

---

> > ### Author Response · Authors · 2024-12-04
> > **Response to Reviewer uwxh**
> >
> > Here’s the results for the new language pair, i.e. en->zh:
> >
> > | Model | Submissions | References | Metric-X (↓) | COMET (↑) | Bleurt (↑) |
> > |---|---|---|---|---|---|
> > |SFT | ✓ | ✓ |  2.215 | 0.536 | 0.700 |
> > |SFT + filter | ✓ | ✓ |   2.220 | 0.531 | 0.695 |
> > |TWA-seq | ✓ | ✓ |    2.172 | 0.540 | 0.701 |
> > |TWA (ignore off-trajectory) | ✓ | ✓ |  2.165 | 0.541 |  0.701 |
> > |TWA (not ignore off-trajectory)| ✓ | ✓ | **2.127**|**0.545**|**0.703**|
> >
> > [Unfortunately, we were not able to run DPO in time for the rebuttal deadline; we will add that baseline to the final draft.]
> >
> > Note that SFT+filter is worse than SFT due to the relatively smaller size of the set of references + perfect submissions for en-zh (compared to the other two language pairs in the draft).
> >
> > Interestingly, for en-zh, similar to zh-en, we observe that including off-trajectory tokens actually helps TWA. As mentioned in the draft and pointed out by the reviewer, the choice of inclusion or exclusion of off-trajectory tokens requires further analysis, and as such we leave this as an interesting direction for future research. We will highlight this observation in Section 4.2 of the final paper.

---

### Official Review · Reviewer_cjx8 · 2024-11-04

**Soundness:** 4
**Presentation:** 4
**Contribution:** 3
**Rating:** 8
**Confidence:** 4

**Summary:**

This work proposes a new method called Training with Annotations (TWA) that leverages the MT evaluation annotation data to improve the quality of machine translation systems. High quality MT evaluation consists of annotation of errors at span-level per example. TWA essentially uses these to annotations to create an additional span level loss while trying to keep
The baselines consist of supervised fine-tuning approaches and DPO based models. The experiments are carried on two language pairs and sufficient ablation studies are conducted.

**Strengths:**

The proposed method is indeed novel - use of MQM annotations to train better MT systems is quite understudied and this work improves on that.

The method looks fairly extensible to other tasks where span level annotations are already available.

The design of the span based loss function carefully considers the potential pitfalls of its inclusion and incorporates additional loss terms to mitigate the same.

**Weaknesses:**

Proposed experiments have been evaluated on high-resource languages. MQM based data is available for Indic languages (https://aclanthology.org/2023.acl-long.795/), African languages (https://aclanthology.org/2024.naacl-long.334/) as well as previous editions of the Quality Estimation Shared Tasks. Evaluation on a mix of different resourced languages can strengthen the contribution of this work.

Not a serious concern with this regards to the content of this work but proposed method is extensible to language pairs/tasks where such annotated data is already available. Future work could indicate potential ways of including synthetic data/alternatives when such high quality annotations are not available.

**Questions:**

Questions:
1. What was the motivation to include a DPO baseline?
2. [Clarification] in the SFT baseline, does the finetuning of the base model involve training with MT triples from the MQM data (without annotations)?
3. Were there any discussions about evaluation on span-based MT metrics like XCOMET (https://arxiv.org/abs/2310.10482) or GEMBA MQM (https://arxiv.org/abs/2310.13988)?



Suggestions:
1. Please include a few more qualitative examples in the Appendix.
2. Please release the code/path to corresponding data after the process.
3. While there is still no consensus about the quality of translations produced by LLMs, it would be useful to add a comment about the extension of this work to LLMs in the discussion section.
4. To get an idea of the effectiveness of this work with contemporary works, it may be useful to report the performance of a few MT models submitted to the WMT'23 shared tasks (where the outputs are already available)

---

> ### Author Response · Authors · 2024-11-27
> **Author Response to Reviewer cjx8**
>
> Thank you for your review and for appreciating the novelty and extensibility of this method! We’ve addressed your questions and suggestions below:
>
> Questions:
> 1. [What was the motivation to include a DPO baseline?]
> - We included DPO as a baseline due to its increasing popularity as a post-training method. DPO represents a baseline that translates MQM information into pairwise comparisons via  sequence-level information only, in contrast to TWA which utilizes the fine-grained MQM annotations directly.
> 2. [Clarification: in the SFT baseline, does the finetuning of the base model involve training with MT triples from the MQM data (without annotations)?]
> - Correct, SFT ignores the annotation information and only trains on the source-target pairs.
> 3. [Were there any discussions about evaluation on span-based MT metrics like XCOMET (https://arxiv.org/abs/2310.10482) or GEMBA MQM (https://arxiv.org/abs/2310.13988)?]
> - We did not consider span-based MT metrics but agree with the author that including these more widely in MT evaluations would be useful!
>
> Suggestions:
> 1. [Please include a few more qualitative examples in the Appendix.]
> - Great suggestion, we’ve added to the appendix some examples from the Zh->En experiments between TWA, SFT, and DPO.
> 2. [Please release the code/path to corresponding data after the process.]
> - Thanks for the suggestion; we are unfortunately not allowed to release the code, but the MQM data is publicly available here, and the pretraining and evaluation data can be found at the respective WMT webpage, e.g. https://www.statmt.org/wmt20/translation-task.html.
> 3. [While there is still no consensus about the quality of translations produced by LLMs, it would be useful to add a comment about the extension of this work to LLMs in the discussion section.]
> - Thanks for the suggestion. We’ve done so in the updated draft.
> 4. [To get an idea of the effectiveness of this work with contemporary works, it may be useful to report the performance of a few MT models submitted to the WMT'23 shared tasks (where the outputs are already available)]
> - As we started with a base model pretrained only on the WMT’23 training data, we do not expect to beat the state-of-the-art submissions to the WMT'23 shared tasks without additional efforts to utilize other data, etc. However, we are excited about future work to incorporate TWA into efforts to achieve state-of-the-art MT models.
>
> We also agree that evaluating on low-resource languages would be really impactful future work, as would considering extensions with synthetic annotations. In preliminary results with a synthetic annotator model (as described Appendix C) we saw positive results, but more thorough investigations would be useful. Thanks again for all the great questions and suggestions!

---

> > ### Comment · Reviewer_cjx8 · 2024-12-03
> >
> > Thank you for the clarifications - I will leave it to the authors to include the results of other contemporary methods in their next iteration.

---

### Author Response · Authors · 2024-11-27
**Overall response**

Thank you to all the reviewers for taking the time to review our work! In response to your comments and suggestions, we:
1. Compared TWA’s span-level loss to a per-token negative loss.
2. Incorporated additional discussion (e.g., significance results, baseline comparisons, negative loss)
3. Included translation examples in the appendix for qualitative analysis.
4. Added experiments for on-policy training.

We’ve responded to each reviewer separately below. We look forward to engaging in additional discussion if anyone has any remaining questions.

---

### Meta-Review · Area_Chair_3JyQ · 2024-12-20

**Metareview:**

This paper introduces a novel method, Training with Annotations (TWA), that leverages machine quality measurement (MQM) annotation data to improve machine translation systems. Specifically, it uses span-level error annotations to create fine-grained supervision via an additional loss term during the fine-tuning process. Experimental results demonstrate improvements over supervised fine-tuning (SFT) and Direct Preference Optimization (DPO) baselines on at least two language pair tasks.

While the paper delivers a compelling contribution with novel methodological implications and demonstrates efficacy in its approach, the paper can benefit from further empirical validation involving more varied language pairs and broader metric justifications. Consider evaluating on a wider range of languages beyond high-resource pairs to demonstrate the model's full potential. Clarifying the implementation of TWA beyond the provided analysis could also strengthen the methodology. The authors can also enhance the baseline conditions by exploring more variations, potentially incorporating suggestions (e.g., weighted contrastive learning, using erroneous examples, and RL-based baselines).

**Additional Comments On Reviewer Discussion:**

During the rebuttal period, reviewers discussed the novelty of TWA in comparison to baseline methodologies and its applicability to low-resource languages. Reviewers also focused on the experiment design, particularly regarding the MQM's data selection and representation. There was a detailed exchange between Reviewer uwxh and the authors, after which Reviewer uwxh stated in the AE-Reviewer discussion, "Although the authors provided some experimental results for new language pairs to address my main remaining concerns, the reported results are limited to a single new translation direction and demonstrate only a very marginal improvement. This improvement appears to result from a potentially selective reporting approach, which undermines their reliability. Therefore, I cannot trust these new results and have decided to maintain my original evaluation."

In the AE-Reviewer discussion, Reviewer cjx8 noted, "My positive score was informed by their design of the method. Span-level annotations have been part of the MT community for a few years, yet it has been challenging to make them work effectively to improve translation methods. However, I agree with the other reviewers regarding (i) the lack of sufficient baselines and (ii) the limited number of language pairs. Additionally, the decision not to release the source code exacerbates the reproducibility issue."

Ultimately, we decided to reject the submission.

---

### Decision · Program_Chairs · 2025-01-22

Reject